# Effectiveness of a reactive oral cholera vaccination during a cholera outbreak at the Douala New-Bell Central Prison in Cameroon

Cavin Epie Bekolo[1]*, Pergui Laure Wetie Tchakoutio[1], Linda Endalle Esso[2], Leonard Ewane[3], Jerome Ateudjieu[1]

1 Department of Public Health, University of Dschang, Dschang, Cameroon, 2 Department for the Control of Disease, Epidemics and Pandemics, Ministry of Public Health, Yaoundé, Cameroon, 3 Littoral Regional Coordination for The Expanded Programme on Immunisation, Ministry of Public Health, Douala, Cameroon

* cavin.bekolo@univ-dschang.org

## Abstract

### Background

Research on cholera vaccine effectiveness in prison populations is scarce, despite evidence of OCV benefits in general outbreak settings. During a major cholera outbreak in Douala's overcrowded central prison, Cameroon deployed the Euvichol oral cholera vaccine. This study assessed the vaccine's association with reduced cholera-related hospitalisation among inmates.

### Methods

A retrospective secondary data analysis was conducted in 2024 on inmates who were targeted for reactive vaccination during a cholera outbreak in 2022. A stratified sampling strategy based on gender and housing type was used to select eligible inmates. Vaccine status was ascertained from the prison's vaccination register. Our outcome variable of interest was cholera-related hospitalisation. Logistic regression was used to assess the likelihood of hospitalisation for severe cholera among vaccinated and unvaccinated inmates. Adjusted odds ratios (aOR) and associated 95% confidence intervals (CI) were obtained.

### Results

A total of 323 inmates were included for analysis, of whom 110 (34.9%) were vaccinated. Cholera-related hospitalisations occurred in 147/213(57.3%) unvaccinated inmates compared to 45/110(41.7%) vaccinated inmates. For inmates who received one dose of OCV, hospitalisation was associated with an adjusted odds ratio of 0.39 (95% CI: 0.21–0.70; p = 0.002), while those who received two doses showed an adjusted OR of 0.37 (95% CI: 0.19–0.72; p = 0.003). In subgroup analyses, severe

**Data availability statement:** All relevant data are within the manuscript and its Supporting Information files.

**Funding:** The author(s) received no specific funding for this work.

**Competing interests:** The authors have declared that no competing interests exist.

cholera was found to be associated with lower odds among female prisoners [aOR = 0.24, 95% CI: 0.08–0.70, p = 0.009] and among inmates in 'VIP' cells [aOR = 0.31, 95% CI: 0.11–0.84, p = 0.022].

## Conclusion

Vaccination was associated with a reduction in hospitalisation during the prison cholera outbreak, confirming its emergency value. However, OCV alone may not be sufficient if overcrowding and poor water, sanitation, and hygiene (WASH) conditions persist. Improved living standards, as seen in the female and VIP sections, must be extended to all inmates.

### Author summary

This study evaluates the effectiveness of a reactive oral cholera vaccine (OCV) in controlling an outbreak at the Douala Central Prison in Cameroon. Cholera, a severe diarrheal disease caused by the bacterium *Vibrio cholerae*, is a major public health concern, particularly in overcrowded and unsanitary environments like prisons. The prison in Douala is a high-risk setting, with severe overcrowding and inadequate sanitation, making it vulnerable to cholera outbreaks.

The study found that inmates who received the OCV had significantly lower chances of being hospitalised due to cholera compared to those who did not receive the vaccine. The results show that even partial vaccination provides meaningful protection, and full vaccination offers the best results. The findings are important because they highlight how vaccines can be a crucial tool in preventing disease outbreaks in prisons, a setting often overlooked in public health strategies. These results not only support the use of vaccines in emergency responses but also emphasise the need for improved sanitation and healthcare access in overcrowded environments. This research contributes to a better understanding of how to manage cholera in high-risk, resource-limited settings.

### Introduction

Cholera, a severe diarrhoeal illness caused by *Vibrio cholerae*, remains a major public health threat in resource-limited settings. Despite progress in water, sanitation, and hygiene (WASH), the disease continues to cause an estimated 1.3 to 4 million cases and up to 143,000 deaths annually [1,2]. To address this burden, the World Health Organization (WHO) launched the "Ending Cholera by 2030" initiative, which calls for strengthened prevention and rapid outbreak response, particularly in vulnerable populations [3].

Among these populations, prisons represent a neglected but critical setting for cholera control. Overcrowding, poor sanitation, restricted mobility, and limited healthcare access create ideal conditions for rapid transmission [4,5]. The Douala Central

Prison in Cameroon illustrates these challenges. Originally built to house 800 inmates, it now accommodates more than 4,000, resulting in extreme congestion and inadequate WASH facilities [5,15]. In April 2022, a cholera outbreak at New Bell Central Prison in Douala, whose attack rate was 5 for 100,000 inhabitants, led to severe dehydration in over 74% of cases and a case fatality rate of 3% [6]. This event highlighted the urgent need for effective containment strategies in correctional facilities.

Reactive oral cholera vaccination (OCV) has emerged as a key intervention in outbreak settings. Over the past decade, vaccines such as Shanchol and Euvichol-Plus have been deployed in refugee camps and urban slums, demonstrating substantial effectiveness in reducing cholera incidence and severity [7]. Studies report reductions in incidence of 65–75% and severity by approximately 78% following reactive campaigns [8,9]. These strategies aim to rapidly immunise at-risk populations during outbreaks to halt transmission.

However, the effectiveness of reactive OCV campaigns in prisons remains underexplored. Correctional facilities present unique structural and operational challenges—high population density, restricted movement, and limited healthcare delivery—that may alter both transmission dynamics and vaccine performance [5,10]. Evaluating vaccine impact in such settings is further complicated by the absence of baseline epidemiological data, as campaigns are often implemented during ongoing outbreaks. Operational hurdles, including maintaining cold chain logistics, navigating security restrictions, and ensuring vaccine acceptance, add further complexity [10–12].

This study seeks to address these gaps by conducting a retrospective evaluation of the reactive OCV campaign at Douala Central Prison. By comparing disease outcomes between vaccinated and unvaccinated inmates during and after the outbreak, and adjusting for confounding factors such as age, housing conditions, and health status, the study aims to quantify vaccine-attributable risk reduction. The findings will provide evidence on the feasibility and effectiveness of emergency vaccination in high-risk institutional environments.

To our knowledge, no prior published data exist on OCV effectiveness among incarcerated populations in Africa. Ultimately, this research contributes to a critical evidence base on cholera control in prisons. By systematically assessing the impact of reactive OCV in a severely overcrowded correctional facility, the study will inform future outbreak responses in similar settings. Strengthening cholera preparedness in prisons is not only a public health priority but also a matter of equity and human rights, ensuring that incarcerated populations are not left behind in global efforts to end cholera.

## Materials and methods

### Ethics statement

The study protocol was reviewed and approved by the Littoral Regional Ethics Committee for Research in Humans *N° 2024/009/CE/CRERSH-LITTORAL,* while permission *N° 002/024/AR/MINJUSTICE/DRAPSAG/LT* was duly obtained from the prison administration before data collection began. Individual consent was not necessary because we used existing data, but all personal identifiers were removed or coded to maintain confidentiality.

### Study setting

The research was conducted at New Bell Central Prison, located in Douala, Cameroon's largest city and commercial hub. The city's rapid urban growth, dense population, and infrastructural inadequacies contribute to a heightened vulnerability to communicable diseases like cholera [13,14]. New Bell Central Prison, situated within the New Bell neighbourhood, operates under extreme infrastructural constraints. Originally built in 1932 to accommodate roughly 800 inmates, current estimates suggest it houses between 4,000 and 5,000 individuals, five times its intended capacity [5]. The resulting overcrowding has created a highly congested environment, with inmates sharing small cells and communal spaces under substandard living conditions [15]. The prison's inadequate WASH infrastructure further exacerbates the risk of waterborne disease outbreaks, a problem common across many African correctional institutions [5,16]. Limited access to clean water and sanitation in such settings hampers effective disease prevention and control. In the wake of the April 2022 cholera

outbreak, prison authorities, in collaboration with health authorities, launched a reactive OCV campaign that reached 3989 inmates with the Euvichol-Plus vaccine, achieving a vaccination coverage of approximately 80.3% [6]. The campaign was supplemented by complementary interventions, including environmental disinfection, water quality improvement, and health promotion activities.

This combination of risk factors, including severe overcrowding, inadequate WASH facilities, and logistical constraints, positions Douala Central Prison as a critical case study for assessing the impact of emergency OCV deployment in institutional settings.

## Study design

We employed an analytical, institutional, cross-sectional design that leveraged available administrative, clinical, and vaccination records to compare inmates who received the vaccine with those who did not in relation to cholera-related hospitalisation after vaccination. This design enabled us to estimate the likelihood of being admitted for severe cholera according to vaccine status while adjusting for potential confounders.

## Study population

All inmates present in prison during the outbreak period were eligible, but only prison records for inmates aged ≥ 15 years, present in prison during the vaccination campaign, and continuously incarcerated for ≥ 7 days post-vaccination were included; inmates with missing vaccination or outcome data were excluded. Access to records is restricted, and specific housing sizes are not publicly disclosed due to human rights concerns about overcrowding in this prison. The exposed group were inmates who received at least one dose of oral cholera vaccine dose ≥ 7 days before symptom onset as recorded in the prison health logs. The unexposed group were inmates who, for reasons such as vaccine hesitancy or contraindications, did not receive the vaccine during the campaign despite being present. These individuals served as the comparison group.

## Sampling strategy and sample size calculation

The stratified sampling strategy was used by dividing the prison population into three distinct strata based on housing: men's sections (about 80% of inmates housed in the standard male confinement areas), women's sections (inmates residing in the female housing units), and VIP sections (20% inmates in specialised or privileged accommodations, which have different living conditions and potentially distinct risk profiles). Each of these housing types was expected to exhibit different characteristics, including population density, environmental conditions, and health risks, making it essential to capture their variability in the sample. Using a proportional allocation method, the total sample size was divided among the strata based on their relative sizes.

Considering an odds ratio of 0.22 for the vaccinated group, which implies an expected incidence of about 2.4% compared to 10% in the unvaccinated group, based on studies from South Sudan and Guinea [8,17]. The sample size calculation using a 1:1 allocation yields approximately 154 inmates per group. Thus, a total of around 308 participants were required for this study to detect the specified difference with 80% power at a 5% significance level.

## Data collection

In this retrospective data collection conducted between March and June 2024, our primary data sources were the existing records maintained by the prison health services and administrative systems, including:

Vaccination registers: these logs provided the dates of vaccine administration, the number of doses given, and the identification of inmate recipients. They were used to classify individuals as "exposed" (vaccinated) or "unexposed" (not vaccinated).

Medical and outbreak records: clinical records, daily health logs, and outbreak surveillance reports maintained during and immediately after the outbreak served to identify cases. These records included the date of diagnosis, clinical

presentation and severity categorisation, laboratory confirmation where available and information on treatment and outcomes. Laboratory confirmation data were available for eleven cases, of whom nine were confirmed positive to *Vibrio cholerae* 01 type by stool culture at the Laquintinie Reference Laboratory. The remaining cases were classified based on WHO clinical criteria used during the outbreak response. Because the outbreak occurred under emergency conditions with operational constraints in the prison infirmary, systematic laboratory testing was not feasible for all suspected cases. As a result, a proportion of clinically diagnosed cases may have included severe non-cholera diarrhoeal illnesses.

Prison administrative records: these records supplied inmate demographics (age, gender, housing unit, duration of incarceration) and movement data. They were essential for verifying eligibility, stratifying the sample, and adjusting for potential confounding factors.

Data extraction and instruments: a standardised electronic data extraction form was developed to capture:

Exposure variables: vaccine status (yes/no), date of vaccination, vaccine dose information.

Outcome variables: date of symptom onset, severity indicators, and outcomes (recovery, hospitalisation, or death). The main outcome variable was hospitalisation, defined as ≥24 hours of admission for intravenous rehydration or clinician determination of severe dehydration.

Covariates: demographics, housing unit, length of incarceration, dates of entry/exit, and other known risk factors (e.g., underlying health conditions, sanitation exposure).

Trained data abstractors: a team of trained personnel extracted data from physical records. By piloting the data extraction tool on a subset of records, we were able to refine the instrument to ensure clarity and consistency. Data verification by cross-checking was performed by comparing information from multiple sources (e.g., vaccination logs and medical records) for selective cases to verify accuracy and completeness.

## Data analysis

Baseline comparisons were done by describing the demographic and clinical characteristics of the vaccinated and unvaccinated groups using means ± standard deviations (for normally continuous variables) or median with their interquartile ranges (for skewed continuous variables), and frequency distributions (for categorical variables). Associations between categorical variables were assessed using Pearson's χ2 test or Fisher's exact test for small samples, as appropriate. For continuous variables, mean differences between the two subgroups were assessed using Student's t-test. This step ensured any imbalances between the groups could be accounted for in subsequent analyses. A logistic regression model where the binary outcome (hospitalisation: yes/no) was modelled as a function of vaccination status and potentially confounding variables such as age, housing section, duration of incarceration, and other relevant covariates. The adjusted odds ratios (OR) for vaccination were obtained and interpreted with their corresponding 95% confidence intervals and p-values. Vaccine effectiveness (VE) was estimated as: VE% = (1−OR) ×100 [16]. To assess the presence of a dose-response relationship between oral cholera vaccine (OCV) doses and hospitalisation, we treated the number of doses (0, 1, 2) as an ordinal variable. A Cochran-Armitage test for trend was performed to evaluate whether the probability of hospitalisation monotonically decreased with increasing OCV doses. Sensitivity analyses were conducted to evaluate the robustness of findings by varying model specifications, reclassifying uncertain cases, or excluding individuals with incomplete data. These analyses helped determine the impact of potential misclassification of exposure or outcome on the effectiveness estimates. All analyses were executed using statistical Stata15.1 (StataCorp LLC, Texas 77845 USA).

## Results

### Study population

A total of 323 inmates were included in the study, comprising 213 unvaccinated individuals (65.9%) and 110(34.1%) who received at least one dose of the oral cholera vaccine (Fig 1). Most participants were males (81.7%), and females

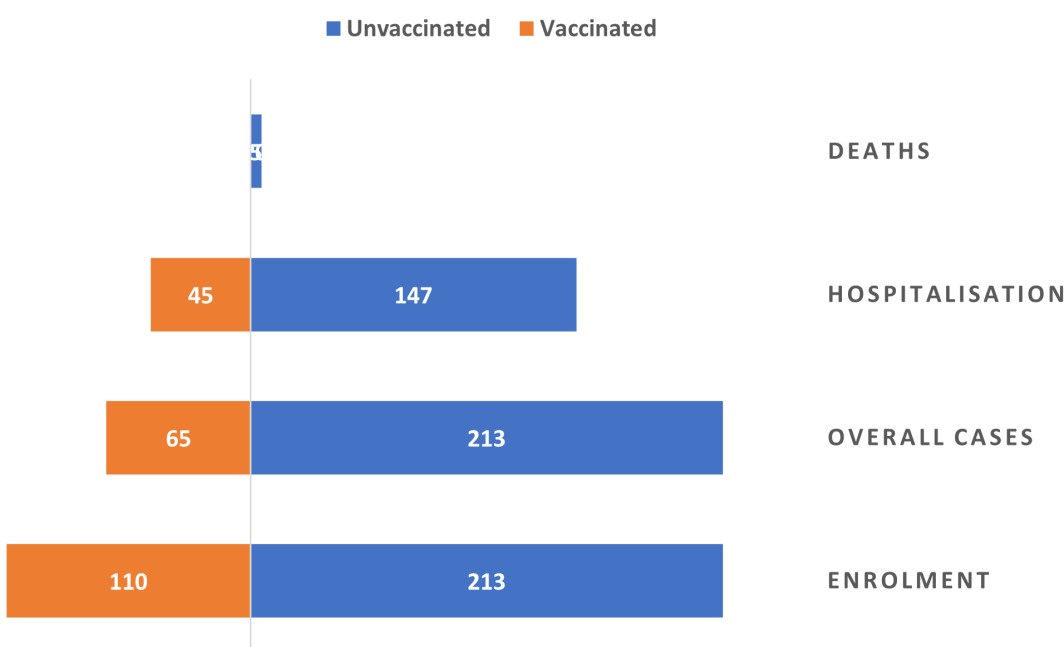

**Fig 1. Flow diagram of record selection and inclusion in the cross-sectional study.**

constituted 18.3%. The median age was 28 years (IQR: 23–34), and only 5.3% of the population were adolescents (<20 years). There was a statistically significant difference in gender distribution between the vaccinated and unvaccinated groups (p < 0.001), with females more likely to be vaccinated (Table 1). There were no statistically significant differences in age group (p = 0.524) or inmate status (awaiting trial vs. convicted, p = 0.958) between the two groups. Housing conditions showed some variation, with 27.3% of vaccinated inmates residing in VIP cells compared to 21.1% of the unvaccinated, although this difference was not statistically significant (p = 0.215).

Clinical presentation differed markedly between groups. Among vaccinated inmates, 94 (85.5%) presented with cholera symptoms early (with mild symptoms), whereas only 73 (34.3%) of the unvaccinated did so. Conversely, 140 (65.7%) of unvaccinated individuals exhibited late presentation, which was significantly higher than among vaccinated inmates (p < 0.001). Early presentation was associated with a reduced need for hospitalisation.

**Effect of oral cholera vaccine on cholera-associated hospitalisation**

Cholera-related hospitalisations occurred in 147/213(57.3%) unvaccinated inmates compared to 45/110(41.7%) vaccinated inmates. The crude odds ratio for hospitalisation among vaccinated inmates was 0.32 (95% CI: 0.17–0.61; p = 0.001), indicating a 68% reduction in odds of hospitalisation. After adjusting for potential confounders (gender, housing unit, and inmate status), the adjusted odds ratio (aOR) was 0.37 (95% CI: 0.19–0.72; p = 0.003). This translated to a vaccine effectiveness (VE) of 63% (Table 2).

A dose-response relationship was also observed. For inmates receiving one dose of OCV, the adjusted odds of hospitalisation were 0.39 (95% CI: 0.21–0.70; p = 0.002), while those who received two doses had an adjusted OR of 0.37 (95% CI: 0.19–0.72; p = 0.003). These figures suggest that while both one and two doses offer protection, complete vaccination is slightly more effective. The trend test showed no significant departure from a linear trend ($\chi^2 = 2.58$, p = 0.11), supporting a monotonic protective effect of OCV across increasing doses (Fig 2).

**Table 1. Baseline characteristics by vaccination status of inmates at the Douala Central Prison.**

| Variables | Unvaccinated group, N = 213 n (%) | Vaccinated group, N = 110 n (%) | p-value |
|---|---|---|---|
| Gender | | | <0.001 |
| Female | 26 (12.21) | 32 (29.09) | |
| Male | 187 (87.79) | 78 (70.91) | |
| Age (years) | | | 0.524 |
| < 20 (Adolescents) | 10 (4.69) | 7 (6.36) | |
| >= 20 (Adults) | 203 (95.31) | 103 (93.64) | |
| Inmate status | | | 0.958 |
| Awaiting trial | 44 (20.66) | 23 (20.91) | |
| Convicted | 169 (79.34) | 87 (79.09) | |
| Housing unit | | | 0.215 |
| VIP cells | 45 (21.13) | 30 (27.27) | |
| Ordinary cells | 168 (78.87) | 80 (72.73) | |
| Clinical presentation | | | <0.001 |
| Early | 73 (34.27) | 94 (85.45) | |
| Late | 140 (65.73) | 16 (14.55) | |

**Table 2. Risk of hospitalisation by OCV dose among inmates of the Douala Prison, Cameroon.**

| Variables | Hospitalisations n (%) | Crude odds ratio OR (95%CI) | p-value | Adjusted odds ratio OR (95%CI) | p-value |
|---|---|---|---|---|---|
| Oral cholera vaccine doses | | | | | |
| 0 | 63 (57.27) | 1 | | 1 | |
| 1 | 27 (43.55) | 0.35 (0.19 – 0.62) | <0.001 | 0.39 (0.21 – 0.70) | 0.002 |
| 1 or 2 | 47 (42.73) | 0.33 (0.21 – 0.54) | <0.001 | 0.38 (0.23 – 0.62) | <0.001 |
| 2 | 20 (41.67) | 0.32 (0.17 – 0.61) | 0.001 | 0.37 (0.19 – 0.72) | 0.003 |
| Gender | | | | | |
| Male | 167 (63.02) | 1 | | 1 | |
| Female | 98 (36.98) | 0.51 (0.29 – 0.91) | 0.022 | 0.24 (0.08 – 0.70) | 0.009 |
| Prison Unit | | | | | |
| Non-VIP | 152 (61.33) | 1 | | 1 | |
| VIP | 45 (59.68) | 0.93 (0.55 – 1.58) | 0.798 | 0.31 (0.11 – 0.85) | 0.022 |

Subgroup analyses revealed significant protective effects by gender and housing status. Female inmates had a notably lower adjusted odds of hospitalisation (aOR = 0.24; 95% CI: 0.08–0.70; p = 0.009) compared to their male counterparts. Similarly, VIP cell residents had lower odds of hospitalisation (aOR = 0.31; 95% CI: 0.11–0.85; p = 0.022) compared to inmates in ordinary cells.

These findings collectively indicated that the reactive OCV campaign significantly reduced the severity and need for hospitalisation among vaccinated inmates during the outbreak.

## Discussion

This study found that during the 2022 cholera outbreak at Douala Central Prison, receipt of OCV was associated with lower odds of hospitalisation and severe disease among inmates. These findings are consistent with prior research from refugee camps and other high-density settings [2,8,11,18] where reactive OCV use has similarly been linked to reductions in cholera-related morbidity. Programmatically, these observations suggest that reactive OCV may offer meaningful

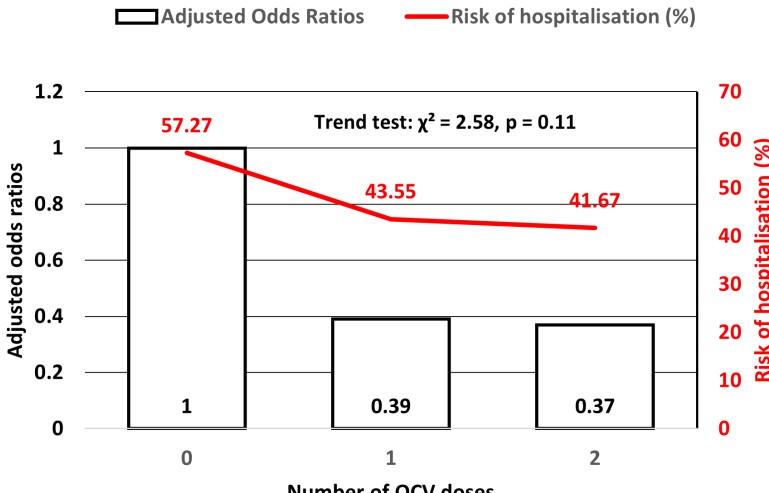

**Fig 2. Cholera-related hospitalisation by oral cholera vaccine dose during the 2022 outbreak at Douala New-Bell Central Prison.**

benefits when integrated into outbreak response strategies for prisons and comparable overcrowded, resource-constrained environments.

A notable strength of this study is the observed dose-response relationship: inmates receiving two OCV doses exhibited marginally greater protection compared to those receiving a single dose. This pattern echoes observations from South Sudan, Bangladesh, and Haiti, where two-dose regimens confer more durable immunity and enhanced protection against severe disease [2,8,18]. However, even a single dose delivered important short-term protection, which is vital in emergency contexts, where completing two-dose schedules can be challenging [8].

Subgroup analysis showed female inmates were more likely to be vaccinated and benefited from a greater reduction in hospitalisation. This may reflect targeted mobilisation efforts or differential health-seeking behaviours among women [19]. Additionally, inmates housed in VIP sections, which offer better sanitary conditions, experienced improved outcomes, highlighting the influence of structural factors such as sanitation and living space on intervention effectiveness and disease risk [20,21]. Therefore, improvements in WASH conditions during the outbreak, particularly in the female and VIP housing blocks where sanitation upgrades were more likely to be implemented, may also have contributed to the lower odds of severe disease observed in these subgroups, and could therefore confound the associations attributed to vaccination.

Our findings should be interpreted considering several key limitations. The sampling frame was constrained by the retrospective nature of the study. Only inmates with complete records for vaccination status and clinical outcomes could be included, which may introduce selection bias if missing data were not random. Although proportional stratification was used to preserve representation of different housing conditions, the final sample reflects the subset of inmates for whom adequate documentation existed, rather than a prospectively defined cohort. The relatively small sample size may have reduced statistical power and the precision of our VE estimates.

Reliance on secondary prison medical and administrative records, which may contain inaccuracies, incomplete entries, or misclassifications that could introduce information bias. Meanwhile, incomplete microbiological confirmation of cases may have affected the precision of our case definitions. Only 5% of cases had laboratory confirmation of Vibrio cholerae by rapid diagnostic testing or culture, while the remainder met WHO clinical criteria for suspected cholera. Although this approach is consistent with outbreak management practices, reliance on clinical diagnosis alone raises the possibility that some cases were due to other severe diarrhoeal pathogens such as ETEC, Shigella, or rotavirus. Such misclassification could bias vaccine effectiveness estimates in either direction—by inflating VE if non-cholera cases occurred disproportionately among

the unvaccinated, or by diluting VE if these cases were evenly distributed across exposure groups. Because cholera was epidemic at the time, we expect most clinically diagnosed cases were true cholera; however, the lack of universal laboratory confirmation should be considered when interpreting the findings. The unbalanced allocation slightly reduced power compared to the planned 1:1 ratio but remained within acceptable margins for detecting the effect size. Unmeasured factors such as nutritional status, prior cholera exposure, or co-infections could also confound the observed associations. The temporal sequence of events between vaccination and the occurrence of cholera could not be ascertained. This period was critical to capture the window during which vaccine-induced protection is expected to manifest, and cholera cases are recorded. Reverse causality and immortal-time bias represent important limitations in our evaluation. In reactive campaigns conducted during an ongoing outbreak, individuals who are already symptomatic or in the incubation period may be less likely to receive vaccination, either because they are too ill to participate or because they were excluded by vaccination teams. This mechanism could spuriously increase the apparent vaccine effectiveness. Although a sensitivity analysis excluding cases occurring within 7 or 14 days after vaccination would normally help assess the robustness of the estimates, this analysis could not be performed because the exact timing between vaccination and symptom onset was not consistently recorded in the available data. As a result, we cannot definitively establish the temporal sequence between exposure and outcome for all participants, and our findings should therefore be interpreted with appropriate caution."

Despite these constraints, our study has important public health implications. Incarcerated populations share many vulnerabilities seen in refugee settings and informal settlements, where WASH services are often inadequate. In such contexts, cholera can spread explosively without swift intervention. OCV serves as a scalable, rapid-response tool to curb outbreaks, particularly when integrated with WASH improvements and targeted health education. Our results support WHO guidance advocating for reactive OCV campaigns within national cholera control frameworks, especially in high-risk, closed environments like prisons [3]. Establishing vaccine stockpiles and pre-approved response protocols can streamline emergency deployments, ensuring timely and equitable access. From a One Health perspective, these findings illustrate how institutional outbreak control enhances health system resilience in vulnerable, densely populated environments. Strengthening health systems through timely vaccine deployment and improved WASH infrastructure can reduce cholera risks for inmates and surrounding communities alike [20].

Moving forward, policymakers should incorporate correctional facilities into cholera preparedness plans, with clear guidelines for vaccine delivery under security constraints. Simultaneously, investments in prison WASH infrastructure are essential to bolster the effectiveness of biomedical interventions. Additional research is needed to evaluate long-term protection, cost-effectiveness, and operational feasibility of OCV campaigns in similar settings.

In conclusion, this study offers valuable evidence on the effectiveness of reactive OCV campaigns in prison settings. Even under challenging conditions and limited baseline data, OCV deployment can significantly reduce cholera morbidity and mortality. When integrated with WASH improvements and health equity efforts, OCV can be a cornerstone of cholera outbreak response in institutional and emergency contexts.

## Supporting information

**S1 File. OCV in the Douala New Bell prison database.**
(PDF)

**S2 File. OCV in the Douala New Bell prison Stata dofile.**
(PDF)

## Acknowledgments

We would like to express our sincere gratitude to the prison administration of Douala Central Prison and the infirmary staff for their invaluable support and cooperation in facilitating this study.

## Author contributions

**Conceptualization:** Cavin Epie Bekolo, Pergui Laure Wetie Tchakoutio, Jerome Ateudjieu.

**Data curation:** Pergui Laure Wetie Tchakoutio.

**Formal analysis:** Cavin Epie Bekolo.

**Investigation:** Pergui Laure Wetie Tchakoutio, Leonard Ewane.

**Methodology:** Cavin Epie Bekolo, Jerome Ateudjieu.

**Project administration:** Linda Endalle Esso, Leonard Ewane, Jerome Ateudjieu.

**Supervision:** Cavin Epie Bekolo, Linda Endalle Esso, Leonard Ewane, Jerome Ateudjieu.

**Validation:** Pergui Laure Wetie Tchakoutio, Linda Endalle Esso, Leonard Ewane.

**Visualization:** Cavin Epie Bekolo.

**Writing – original draft:** Cavin Epie Bekolo, Pergui Laure Wetie Tchakoutio.

**Writing – review & editing:** Cavin Epie Bekolo, Linda Endalle Esso, Jerome Ateudjieu.

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
