## [Decision Letter · Decision Letter 0]

19 Oct 2025

Effectiveness of a Reactive Oral Cholera Vaccination during a cholera outbreak at the Douala New-Bell Central Prison in Cameroon

Dear Dr. Bekolo,

Thank you for submitting your manuscript to PLOS Neglected Tropical Diseases. After careful consideration, we feel that it has merit but does not fully meet PLOS Neglected Tropical Diseases's publication criteria as it currently stands. Therefore, we invite you to submit a revised version of the manuscript that addresses the points raised during the review process.

Please submit your revised manuscript within 60 days Dec 18 2025 11:59PM. If you will need more time than this to complete your revisions, please reply to this message or contact the journal office at plosntds@plos.org. Please include the following items when submitting your revised manuscript:

We look forward to receiving your revised manuscript.

Kind regards,

Elsio A Wunder Jr, DVM, Ph.D.

Section Editor

Elsio Wunder Jr

Section Editor

Shaden Kamhawi

co-Editor-in-Chief

Paul Brindley

co-Editor-in-Chief

**Additional Editor Comments (if provided):**

**Journal Requirements:**

**Reviewers' Comments:**

Reviewer's Responses to Questions

**Key Review Criteria Required for Acceptance?**

**Methods**

-Are the objectives of the study clearly articulated with a clear testable hypothesis stated?

-Is the study design appropriate to address the stated objectives?

-Is the population clearly described and appropriate for the hypothesis being tested?

-Is the sample size sufficient to ensure adequate power to address the hypothesis being tested?

-Were correct statistical analysis used to support conclusions?

-Are there concerns about ethical or regulatory requirements being met?

Reviewer #1: The study's objective is clear, but a formal hypothesis is not stated. The retrospective design is appropriate but inherently vulnerable to temporal bias. The prison population is well-defined and relevant. However, the sampling methodology and justification for the final sample size (n=323) require clearer articulation. The statistical analyses are generally appropriate, but sensitivity analyses are needed to confirm the robustness of the primary findings against potential confounding and reverse causality.

Reviewer #2: The methods are clearly explained and presented.

Reviewer #3: a. Study design and population

• Line 144–151: The term “institutional cross-sectional study” contradicts causal inference, relabel as retrospective cohort or analytic cross-sectional depending on temporal data.

• Provide inclusion/exclusion criteria (age ≥ ? ; duration of incarceration > ? ).

b. Sampling

• Lines 156–165: Stratified sampling by gender/housing type, state exact strata sizes and sampling fractions.

c. Exposure definition

• Lines 93–100 & 172–179: Specify vaccine product (Euvichol® vs Euvichol-Plus®), dose interval, and campaign dates; define “vaccinated” as ≥ 7 days post-dose.

d. Outcome definition

• Lines 179–193: Define “cholera-related hospitalisation”: was it based on IV rehydration, ≥ 24 h stay, or clinician judgment?

f. Sample-size calculation

• Lines 166–171: Discuss how realised allocation (~2:1 unvaccinated:vaccinated) affects power.

**Results**

-Does the analysis presented match the analysis plan?

-Are the results clearly and completely presented?

-Are the figures (Tables, Images) of sufficient quality for clarity?

Reviewer #1: The analysis follows the planned approach to use logistic regression. It provides both crude and adjusted odds ratios for the main factor, vaccination, and important subgroups. However, the manuscript does not include the sensitivity analyses mentioned in the methods. These analyses are essential for testing how reliable the findings are against potential biases like reverse causality.

The results for vaccine effectiveness are clearly stated. A major gap is the failure to report the proportion of "cholera-related hospitalizations" that were confirmed by a lab. This omission makes it hard to evaluate potential misclassification of outcomes. Additionally, the results in Table 2 are presented in a confusing and probably incorrect way. The column labeled "Number of hospitalizations n (%)" seems to list the total number of subjects in each subgroup instead of the number actually hospitalized, which reduces the clarity of the data.

The single table provided is basic and does not offer complete clarity. Table 1 is adequate, but Table 2, as mentioned, needs immediate correction of its headers and data presentation to make it understandable. The biggest issue is the lack of a participant flow diagram, such as a CONSORT-style diagram. This is a standard tool for clarifying the sampling process from the total prison population to the final analytical sample. Without it, the selection process remains unclear.

Reviewer #2: The results are clearly presented and are logical.

Reviewer #3: a. Descriptive statistics

• Lines 228–237: Only 34.9 % vaccinated, though campaign coverage was > 80 %, explain discrepancy (missing records or population turnover).

• Provide a flow diagram: eligible → sampled → analysed.

b. Table 2 (lines 255–266)

• Denominators and percentages inconsistent (e.g., 20/110 ≠ 41.7 %). Recalculate and show n/N (%) for each dose level.

• Separate baseline characteristics (Table 1) from outcome (Table 2).

• Add absolute risk difference and risk ratio alongside ORs.

• Add p-trend test for 0, 1, 2 doses.

c. Dose–response claim

• Lines 264–268: The aORs (0.39 vs 0.37) overlap; do not assert a gradient without statistical test.

e. Figures

• Add epidemic curve (onset dates) with vaccination timeline to confirm exposure precedes outcomes.

**Conclusions**

-Are the conclusions supported by the data presented?

-Are the limitations of analysis clearly described?

-Do the authors discuss how these data can be helpful to advance our understanding of the topic under study?

-Is public health relevance addressed?

Reviewer #1: The conclusions are supported by the primary data, demonstrating a clear protective effect of vaccination. However, the discussion of limitations is insufficient, particularly regarding the critical risk of reverse causality and the lack of lab confirmation for all cases. The authors excellently advance understanding by highlighting the vaccine's utility in a neglected prison setting and its synergy with improved living conditions. The public health relevance is strongly and effectively addressed, with direct implications for outbreak response in institutional settings.

Reviewer #2: Yes, the conclusions are supported by the data presented.

Reviewer #3: (No Response)

**Editorial and Data Presentation Modifications?**

Reviewer #1: This study examines the effectiveness of OCV in a neglected prison population. The main finding, that vaccination lowers hospitalization rates, is clear and significant. However, significant revisions are needed to strengthen its validity. The analysis should tackle key limitations, such as the possibility of reverse causality, which means sicker inmates may not have received vaccines. This should be explored through a sensitivity analysis. The study also lacks clarity on lab-confirmed cases. Additionally, the sampling method requires a clearer explanation, and a flow diagram would be helpful. Table 2 needs correction because its headers can be misleading. The assertion that two doses are "slightly more effective" lacks support from the overlapping confidence intervals and should be reworded. Addressing these concerns is crucial for publication.

Reviewer #2: None - see summary for minor comments

Reviewer #3: Tables, Figures, and Supplementary Files (pp. 13–16)

• Tables: Ensure denominators consistent; add clear titles (“Baseline characteristics by vaccination status,” “Risk of hospitalisation by OCV dose”).

• Figures: Epidemic curve and dose-response plot recommended.

• Supporting data

**Summary and General Comments**

Reviewer #1: This study provides important evidence on the effectiveness of reactive oral cholera vaccination in a prison setting, which is a poorly studied and vulnerable population. The study is well-motivated, and its findings are relevant for public health policy. The methods used are mostly suitable. However, major revisions are needed to address key limitations. Required analyses include: 1) a sensitivity analysis to eliminate reverse causality bias, such as excluding cases that occur shortly after vaccination, and 2) clarification on the percentage of lab-confirmed cholera cases. The sampling strategy and the unclear presentation of results in Table 2 also need correction. Once these issues are resolved, this will make a strong contribution to the field.

Reviewer #2: In this manuscript, Bekolo and colleagues report the results of a retrospective analysis of OCV effectiveness from a reactive vaccination campaign in a correctional facility in Cameroon. They find that OCV administration was significantly associated with a lower risk of hospitalization in a dose-dependent manner, and identify other contributing factors that may be useful to consider when designing vaccination campaigns for high-density populations. Although this reviewer is not an expert in clinical trial analysis, I find the data analysis and conclusions sound and the authors have carefully limited their extrapolation. This is a well-written, clear, and valuable study that draws a useful comparison between prison settings and other high-density populations that require effective vaccination campaigns, such as refugee camps or densely settled neighborhoods.

- The timeline of the outbreak (April 2022) in relation to the vaccination campaign is not clear - could the authors provide a visual timeline or explain this in the text? This could well have impacted vaccine efficacy measurements.

- The rate of hospitalization in the overall population was very high, and the authors were only looking at data for severe disease in Table 2. Although there is some analysis in Table 1 of vaccinated vs. unvaccinated clinical presentation, do the authors have data on the overall incidence of cholera-like illness (i.e., how many inmates had no clinical presentation)?

- Were any of the cases culture-confirmed to be V. cholerae infections?

Reviewer #3: 1. Abstract (pp. 2–3, lines 25–51)

• Clarity of design: The phrase “retrospective secondary data analysis” could mislead reviewers, since effectiveness implies temporality. Clarify whether this is a retrospective cohort or cross-sectional analytic design.

• Wording: Correct “adjusted odd ratios” → “adjusted odds ratios.”

• Data transparency: Add denominators for vaccinated/unvaccinated participants (e.g., “20/110 vs 63/213 hospitalised”).

• Interpretation: Avoid causal language (“significantly reduce”) until temporality bias is addressed; use “associated with lower odds.”

• Line 45–46: The conclusion “may not be enough to control cholera” should be tempered—tie to concurrent WASH improvements.

2. Introduction (pp. 3–4, approx. lines 52–105)

• The background establishes relevance but omits literature on prison settings. add one or two prior OCV studies in confined institutions.

• Add a quantitative statement of global cholera burden to frame importance (e.g., WHO 2024 est.).

• Clarify research gap: “No prior published data on OCV effectiveness among incarcerated populations in Africa.”

PLOS authors have the option to publish the peer review history of their article (what does this mean? ). If published, this will include your full peer review and any attached files.

**Do you want your identity to be public for this peer review?** For information about this choice, including consent withdrawal, please see our Privacy Policy .

Reviewer #1: No

Reviewer #2: No

Reviewer #3: **Yes: ** Charles Lwanga Noora

**Figure resubmission:**
---

## [Decision Letter · Decision Letter 1]

17 Dec 2025

Dear Dr. Bekolo,

We are pleased to inform you that your manuscript 'Effectiveness of a Reactive Oral Cholera Vaccination during a cholera outbreak at the Douala New-Bell Central Prison in Cameroon' has been provisionally accepted for publication in PLOS Neglected Tropical Diseases.

Best regards,

Elsio A Wunder Jr, DVM, Ph.D.

Section Editor

Elsio Wunder Jr

Section Editor

Shaden Kamhawi

co-Editor-in-Chief

Paul Brindley

co-Editor-in-Chief

Reviewer's Responses to Questions

**Key Review Criteria Required for Acceptance?**

**Methods**

-Are the objectives of the study clearly articulated with a clear testable hypothesis stated?

-Is the study design appropriate to address the stated objectives?

-Is the population clearly described and appropriate for the hypothesis being tested?

-Is the sample size sufficient to ensure adequate power to address the hypothesis being tested?

-Were correct statistical analysis used to support conclusions?

-Are there concerns about ethical or regulatory requirements being met?

Reviewer #2: (No Response)

**Results**

-Does the analysis presented match the analysis plan?

-Are the results clearly and completely presented?

-Are the figures (Tables, Images) of sufficient quality for clarity?

Reviewer #2: (No Response)

**Conclusions**

-Are the conclusions supported by the data presented?

-Are the limitations of analysis clearly described?

-Do the authors discuss how these data can be helpful to advance our understanding of the topic under study?

-Is public health relevance addressed?

Reviewer #2: (No Response)

**Editorial and Data Presentation Modifications?**

Reviewer #2: (No Response)

**Summary and General Comments**

Reviewer #2: The authors have satisfactorily addressed my comments.

PLOS authors have the option to publish the peer review history of their article (what does this mean? ). If published, this will include your full peer review and any attached files.

**Do you want your identity to be public for this peer review?** For information about this choice, including consent withdrawal, please see our Privacy Policy .

Reviewer #2: No

---

## [Editor Report · Acceptance letter]

Dear Dr. Bekolo,

We are delighted to inform you that your manuscript, "Effectiveness of a Reactive Oral Cholera Vaccination during a cholera outbreak at the Douala New-Bell Central Prison in Cameroon," has been formally accepted for publication in PLOS Neglected Tropical Diseases.

Best regards,

Shaden Kamhawi

co-Editor-in-Chief

Paul Brindley

co-Editor-in-Chief
